# MediaHG: Rethinking Eye-catchy Features in Social Media Headline Generation

**Boning Zhang** and **Yang Yang**[*]

Zhejiang University

zhangbn@zju.edu.cn and yangya@zju.edu.cn

## Abstract

An attractive blog headline on social media platforms can immediately grab readers and trigger more clicks. However, a good headline shall not only contract the main content but also be eye-catchy with domain platform features, which are decided by the website's users and objectives. With effective headlines, bloggers can obtain more site traffic and profits, while readers can have easier access to topics of interest. In this paper, we propose a **disentanglement-based** headline generation model: **MediaHG** (Social **M**edia **H**eadline **G**eneration), which can balance the content and contextual features. Specifically, we first devise a sample module for various document views and generate the corresponding headline candidates. Then, we incorporate contrastive learning and auxiliary multi-task to choose the best domain-suitable headline, according to the disentangled budgets. Besides, our separated processing gains more flexible adaptation for other headline generation tasks with special domain features. Our model is built from the content and headlines of 70k hot posts collected from *REDBook*, a Chinese social media platform for daily sharing. Experimental results with language metrics ROUGE and *human evaluation* show the improvement in the headline generation task for the platform[1].

## 1 Introduction

Nowadays, in the midst of massive flow of information on large-scale social network sites (such as *Facebook* and *Instagram*), users always feel more difficult to quickly obtain the information they want. As a result, people tend to focus on more niche platforms, which are often made up of groups of users with common personal interests. The vertical platforms are not only used for

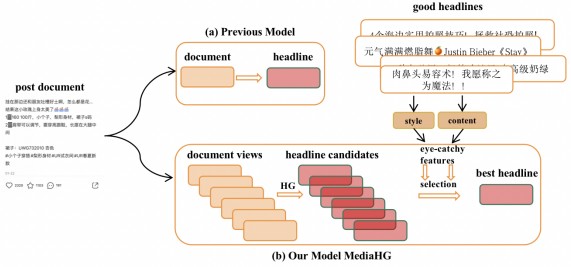

Figure 1: Comparison and improvement with previous work. (a) Previous Seq2Seq models input the whole document to generate a headline, resulting in the omission of the main topic and latent features. (b) In contrast, MediaHG samples document views to capture the main topic and selects the best headline with eye-catchy budgets. Besides, the eye-catchy feature extractor is disentangled with content and style to achieve a duality balance.

easier information search but also strengthen the community of users with the same interests. Users' attention is always limited to attractive headlines that can catch their eyes at first glimpse. As the headline condenses the main topic into a concise and appealing description, a good headline can trigger a high click rate. So generating better headlines is significant for media platforms to compete for attracting users' limited attention and deliver users better experiences.

We conduct the research based on a vertical Chinese social media platform *REDBook* (Figure 2) because it is widely praised by hundreds of millions of users and targeted by the same interest group. As more than 70% of users are female (an official reported data), the topics and tone of posts are feminine such as some typical words "Babycare", "Makeup" etc. The platform breaks down identity restrictions and allows people to share their colorful lives and experience, which are of reference value for users. We define the *eye-catchy* features of posts as the ability to attract users, which can be intuitively measured by the number of "likes",

---

[*]Corresponding author

[1]The code is available at https://github.com/Rosenn2000/MediaHG

that is, the heat of posts on the platform. Producers can obtain more traffic and profits while getting more likes from the platform, then they may be subscribed by more fans. The advertising revenue of bloggers is usually closely related to both the number and profile of their followers. So only with the help of good headlines, can the bloggers attract their target user group.

By analyzing eye-catchy *REDBook* headlines with more likes (more than 2k) on the platform, we find that both contents and style influence attractiveness. Since the majority of users are women, topics of interest to women occupy a large part of the topics on the platform. Accordingly, headlines with domain special topics such as *"Lipstick swatches"* or *"ootd"* ("Outfit of the Day") are more appealing with more likes. Combined with hot topics, the style of the headline also has a great impact on eye-catchy features. For example, when reporting the hot issue "lipsticks sharing", the headline "New Lipsticks for Winter  So Tender like a Creamy Almond Peach!!" (as shown on the platform "冬季口红新品～奶茸茸的杏仁桃子好温柔" in Chinese) wins over 13,000 likes, while another headline with the same topic but a plain description "Lipstick Share, a New Style for Winter" only has 300 likes. We also find the eye-catchy style of the headlines on the platform is not that similar to the style of news media headlines, as it more closely resembles the tone of women talking and sharing with their friends.

Focusing on relevant works, we found that recent researches simply regard the headline generation task as a typical summarization task (Shu et al., 2018). They only focus on the content parallel to the given reference summary, ignoring the domain eye-catchy features. However, attractive headline-generation tasks have received less attention. A recent clickbait research (Xu et al., 2019) leverages adversarial training and attractiveness scores module to guide the summarization process. Another (Jin et al., 2020) introduces a novel parameter-sharing scheme to disentangle the attractive style from the text. However, these previous works only concentrate on style and neglect the content importance, which also weighs in eye-catchy headline generation. Disentanglement module is devised to divide the style and content into latent spaces, but a style encoder in generator training is not flexible enough(Li et al., 2021).

To address the headline generation issue, we propose the MediaHG model which disentangles the eye-catchy features as additional requirements in sequence-to-sequence training. The model is composed of a headline candidates generator and an eye-catchy headline selector. In our setting, the neural abstractive model is responsible for headline generation, capturing the main topic of the input document, while the selection module with constraint will encourage the adherence of generated headlines to domain eye-catchy features. Instead of confounding eye-catchy features, we treat the content feature and style feature extraction respectively. In the generation period, we devise a random sample module with different parts of the text and generate candidate headlines responding to the content. During selection period, we leverage ranking-based contrastive learning(Hopkins and May, 2011)(Zhong et al., 2020)(Liu et al., 2021) and multi-task(Luong et al., 2015) to select the best headlines among candidates. The selection is decided by the coordinating quality scores of style-content attractiveness. We will describe the specific quality metrics model in detail in the following part. Therefore, candidates are assigned with probabilities according to their quality, which will further influence the generation model. In other words, the headline generation model not only generates output headlines autoregressively but also estimates the probability distribution over candidate headlines.

Our main contributions are listed below:

- We propose a new Headline Generation model namely MediaHG to generate topic-catchy and contextual harmonized headlines for vertical niche platforms to enhance the click ratio and draw users' attention. While we base our experiments on a typical vertical platform *REDBook*, our methods can be adapted to other platforms through the same platform-suitable features extraction methods.

- Our model is proved to be effective by both automatic and human evaluation scores of fluency, consistency, and attractiveness, which means it achieves a style-content dual balance.

- To the best of our knowledge, it is the first research to focus on vertical interest platforms. We also give a new definition of domain *eye-catchy* headlines that is, those attractive combinations with topic and style suitable to the platform users.

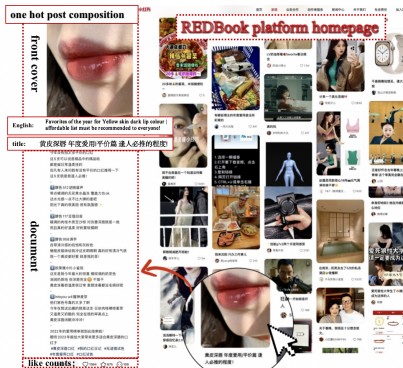

Figure 2: REDBook Homepage and Specific Hot Post Page Display. The layout of the browsing page shows only the posts' cover and the title, thus emphasizing the importance of the title.

## 2 Related work

**Headline Generation** Text generation has undergone impressive progress in recent years (Li et al., 2019)(Chan et al., 2019)(Liu et al., 2020)(Xie et al., 2020)(Chan et al., 2020), and headline generation occupies the dominance. Most existing works merely focus on document summarization with extractive (Nallapati et al., 2017)(Zhou et al., 2018)(Zhang et al., 2018) or abstractive (Gao et al., 2019)(Chen et al., 2019b)models. Several methods regard headline generation as a task based on length-controlled text summarization. Controlling length in summaries has been addressed by encoding positional information (Takase and Okazaki, 2019), a length-aware attention mechanism (Liu et al., 2022), and a length constraint optimization(Makino et al., 2019). Content guidance in GSum(Dou et al., 2020) is used as the input for its sequence-to-sequence model, and shifts in guidance distribution would require further training. Attractive headline generation is paid less attention by researchers. A sensation scorer(Xu et al., 2019) is designed to judge whether a headline is attractive and guide the headline generation by reinforcement learning. Also, a parameter-sharing scheme(Jin et al., 2020) is introduced to further extract style from the text. The style-content duality is considered with *VAE* (Variational AutoEncoder) as a feature extractor (Li et al., 2021) and two disentangled space constraints in parallel tasks. Differently, MediaHG allows flexible shifts in various guidance without expensive retraining.

**Disentanglement** Disentangling neural networks' latent space has been explored in the computer vision domain to factorize the features (such as ro-

tation and color) of images (Chen et al., 2016)(Higgins et al., 2017)(Luan et al., 2017). Compared to the computer vision field, NLP tasks mainly treat sentiment as a salient style and focus on invariant representation learning. It is used to control sentiment through training a discriminator (Hu et al., 2017). Then, disentangled representation learning is further widely adopted in nonparallel text style transfer. For example, separate training with style-specific embeddings and style-specific embeddings are proposed(Fu et al., 2018). Some work also focuses on disentangling syntax and semantic representations in text. VGVAE(Chen et al., 2019a) trains the generative model with multiple losses that exploit aligned paraphrastic sentences and word-order information to get better syntax and semantics representations. We utilize the core principle of disentangling to separate different feature budgets.

**Reranking Candidates** Recent conditional generation work explores the idea of reranking candidates from different dimensions(Wan et al., 2015)(Mizumoto and Matsumoto, 2016). Different search methods have been used in neural language summarization models, such as greedy search in FactorSum (Fonseca et al., 2022) and beam search (Vijayakumar et al., 2016) in SimCLS (Liu and Liu, 2021)according to a learned evaluation function. The Perturb-and-Select summarizer (Oved and Levy, 2021) performs random perturbations and uses similar ideas to generate candidates ranked according to a coherence model. Unlike only intrinsic importance compared with the original document in SimCLS, content and contextual eye-catchy budgets are both considered in our work.

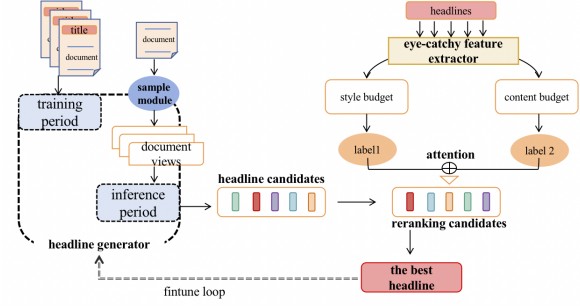

Figure 3: Overview of MediaHG. We divide our models into 3 parts: *Sample Module* for headlines candidates generation; *Headline Generator* with different datasets training and inference; *Features Reranking* to select the domain-best title according to disentangled style and content budgets.

# 3 Approach

In this section, we describe our approach in detail, as shown in Figure 3. Inspired by FAC-TORSUM (Fonseca et al., 2022), we treat the content importance model as sampling document views(intrinsic importance), and contextual features as additional budgets(extrinsic importance). We pre-train the headline generation model with our dataset *REDBook* (Table 1) to generate candidates with intrinsic content and latent features in Sec3.1. Then, a specific metric $\mathcal{M}$ is employed to evaluate the effectiveness of different criteria, composed of scores from both content and domain style factors in Sec3.2. To assign higher probabilities to a more suitable candidate, we use contrastive learning for better re-ranking. The metric $\mathcal{M}$ construction is demonstrated in Sec3.3 which is optimized with multi-task loss.

## 3.1 Generate Candidates

The headline candidates are generated from two tasks: document parts sampling and corresponding headline generation. The candidate document views are generated from different random samples of article parts. We hypothesize that the main topic of short posts (limited to 1000 Chinese characters) shall be contracted from various sampled incomplete parts. Using samples allows the sequence-to-sequence model to focus on concise and appealing topics, as we further considered.

To generate multiple views for the same document, we implement the following steps:

- From a document **D**, we first split the sentences and generate a random sample collection of sentences, called document views $S_v$. The number of sentences of each document view in $S_v$ is controlled by the sampling parameter $s_f \in [0, 1]$, so that $sents(S_v) \approx s_f \cdot sents(D)$. The number of samples $|S_v|$ is controlled with hyperparameter $k$.

- We generate the headline corresponding to each document view in $S_v$ with a pre-trained sequence-to-sequence model. The model defaults to generate headlines with content and eye-catchy features. The headlines collections $\{h_1, h_2...h_{n_d}\}$ cover $n_d$ candidates inside.

For each document in the *RED-IN* (Table 1), we repeat the sampling method and headline generation work. While dealing with different datasets containing various lengths of documents and titles, the hyperparameters may be tuned. According to the basic *REDBook* platform format restrictions, the length of the document and title is limited to 1000 characters and 20 characters respectively. So we choose $k$ candidate document views, each with $s_f = 2/3$ sentences of the document. Different choices with appropriate values $k$ are discussed in the ablation study.

Powerful sequence-to-sequence PLMs models such as PEGASUS(Zhang et al., 2020) and BART (Lewis et al., 2019) are trained to estimate the probability of a sequence of tokens by minimizing cross-entropy with respect to the data distribution. We hypothesize that these models generate good candidates to fulfill the content importance and latent style features objectives, which are described below.

**Learning** Maximum likelihood estimation (MLE) is the standard training algorithm. Given the training dataset $X'$ consists of hot post document $D_{(i)}$ and reference headline $H_{(i)}$, the loss is defined as a negative log-likelihood function:

$$L\big(E_{int}, \mathcal{X}'\big) = \frac{1}{|\mathcal{X}'|} \sum_{i=1}^{|\mathcal{X}'|} \underbrace{L\big(H_{(i)}, E_{int}\big(\theta, D_{(i)}, H\big)\big)}_{-\log p_\theta\big(H_{(i)}|D_{(i)}\big)} \tag{1}$$

where $p_\theta\big(H_{(i)} \mid D_{(i)}\big)$ is a distribution over the possible headline $H$ (Lewis et al., 2019).

For a specific sample $\{D^{(i)}, H^{*(i)}\}$, Eq.2 is equivalent to minimizing the sum of negative log-likelihood of the tokens in the reference headline $H^*$ whose length is $l$, through the cross-entropy loss:

$$\mathcal{L}_{sent} = \tag{2}$$
$$-\sum_{j=1}^{l}\sum_{h} p_{\text{true}}\big(h \mid D, H^*_{<j}\big) \log p_{g_\theta}\big(h \mid D, H^*_{<j}; \theta\big)$$

where $H^*_{<j}$ denotes the partial reference headline $\{h^*_0, \cdots, h^*_{j-1}\}$. $p_{true}$ is defined as a one-hot distribution under the standard MLE framework:

$$p_{\text{true}}\big(h \mid D, H^*_{<j}\big) = \begin{cases} 1 & h = h^*_j \\ 0 & h \neq h^*_j \end{cases} \tag{3}$$

During learning stage, we find the parameters $\theta_*$ minimize the loss above. Since the model is trained with a confounded feature dataset, we notice the

results are generated with both content and latent features.

**Inference** During inference stage, the abstractive model $g$ is used to generate the candidate headline in an autoregressive manner. It is intractable to enumerate all the possible candidate outputs, so methods such as beam search decoding(Sutskever et al., 2014) are used to reduce the search space. Estimating the probability of the next word $h_t$ is the significant step during the search:

$$p_{g_\theta}\left(h_t \mid D, H_{<t}; \theta\right) \qquad (4)$$

which is different from Eq.3 with its own previous predictions resource $H_{<t}$ instead of reference headline $H_{<t}^*$.

## 3.2 Coordinating Headline Selection

Eq.4 implies that the headline generation model $g$ should be able to assign a higher estimated probability to the better candidate summary during inference. However, this intuition is not directly captured in the standard MLE objective used in training. No option is adopted for the ordering of imperfect references, which will lead to the existence of multiple generations(Khayrallah et al., 2020). Therefore, we propose the probability that one candidate should be well-correlated with its quality as evaluated by an automatic feature metric $\mathcal{M}$. It is intractable to enumerate all the possible candidate outputs, so we only require an accurate prediction of the most probable candidate headlines ranking order via beam search (See Appendix A).

We use label-smoothing (Szegedy et al., 2016) and maintain the general functional form Eq.3, but specify the marginal probability of the non-reference candidates $\mathcal{H}$ to be $\beta$. Additionally, we encourage the coordination of probabilities and qualities among headline candidates by contrastive learning as follows:

$$
\begin{cases}
p(H \mid D) = 1 - \beta & H = H^* \\
\sum_{H \in \mathcal{H}} p(H \mid D) = \beta & H \neq H^* \\
p^\dagger(H_i \mid D) > p^\dagger(H_j \mid D) & \forall H_i, H_j \in \mathcal{H}, \\
& M(H_i) > M(H_j)
\end{cases}
\qquad (5)
$$

The candidate quality measure $M$ in our work is defined with two scores: content score and style score which are responsible for the topic extraction and contextual media style features, separately.

$$\mathcal{M}(H_i) = \alpha m_c(H_i) + (1-\alpha)m_s(H_i) \qquad (6)$$

where $m_c$ score measures the topic components of a candidate headline $H_i$ extracted from the document and style score $m_s$ measures the contextual latent features. We fine-tune the model with contrastive loss (Hopkins and May, 2011)(Zhong et al., 2020) which encourages the model to assign higher probabilities to a more suitable candidate as follows:

$$\mathcal{L}_{ctr} = \sum_i \sum_{j>i} \max\left(0, f\left(H_j\right) - f\left(H_i\right) + \lambda_{ij}\right) \qquad (7)$$

where $H_i$ and $H_j$ are two different candidate headlines and $\mathcal{M}\left(H_i, H^*\right) > \mathcal{M}\left(H_j, H^*\right), \forall i, j, i < j$ by metrics $M$. $\lambda_{ij}$ is the margin multiplied by the difference in rank between the candidates, i.e. $\lambda_{ij} = (j - i) * \lambda$.

Following multi-task fine-tuning (Edunov et al., 2017), we combine the contrastive (Eq.7) and cross-entropy (Eq.2) losses to preserve the generation ability of the pre-trained abstractive model:

$$\mathcal{L}_{mul} = \mathcal{L}_{sent} + \gamma \mathcal{L}_{ctr} \qquad (8)$$

where $\gamma$ is the weight of the contrastive loss. We note that the contrastive and the cross-entropy loss can effectively complement each other. Since the contrastive loss is defined on the eye-catchy features, the token-level cross-entropy loss serves as a normalization to ensure content-style balanced probability assignment. This optimization loss of the result can be used in the two-stage summarization pipeline.

## 3.3 Disentangled Space Constraint

The disentanglement scores framework shown in Figure 3 consists of content (intrinsic) constraints and appealing style (extrinsic) constraints.

**Content Space Constraint** As the above style-oriented loss has already imposed constraints on the style information, the content space constraints methods will be discussed in this part. Different from the style constraint design, it is hard to find parallel sentences with the same content but different styles. Previous work DAHG(Li et al., 2021) used the prototype document and its most similar document to improve the classifier. However, the similarity precision is not clearly defined. The bag-of-words (*BOW*) method is proposed to approximate content information(John et al., 2018) disentanglement in document style transfer tasks, but our generation objectives are concise headlines.

Inspired by the original (*BOW*) method, we use ROUGE(Lin, 2004)scores to measure the main content overlapping.

**Style Space Constraint** We design a multi-task loss that ensures the style information is contained in the space $\mathcal{S}$. Although our dataset for style extraction is non-parallel, we assume that each sentence is labeled with its style (with domain eye-catchy features or not). We select the eye-catchy headlines from the platform and other plain corpora sentences to train the style classifier. Following the previous work (Hu et al., 2017)(Shen et al., 2017)(Fu et al., 2018)(Zhao et al., 2018) we treat each sentence with a binary style tag (positive or negative).

To disentangle the style information, two headlines $H^p$ and $H^n$ with different labels are selected as two candidates for the classifier. Then the headlines are embedded with the same matrix to obtain the representation $h^p$ and $h^n$ of $H^p$ and $H^n$, respectively. A two-way softmax layer (equivalent to logistic regression) is applied to the style vector $\int$:

$$y_s\left(H^*\right) = \text{softmax}\left(W_{ss}\left[s; h^*\right] + b_{ss}\right). \quad (9)$$

where $\theta_{mul(s)} = [W_{ss}; b_{ss}]$ are parameters for multi-task learning of style, and $y_s$ is the output of softmax layer. The classifier is trained with a cross-entropy loss against the ground truth distribution $c_s\left(\cdot\right)$, shown as

$$L_{\text{mul}}\left(\boldsymbol{\theta}_{\text{E}}; \boldsymbol{\theta}_{\text{mul}}\right) = -\sum_{H \in \text{labels}} c_s(l) \log y_s(H) \quad (10)$$

The optimization can be viewed as multi-task learning loss at the same time. It not only auto-decodes the sentence but also predicts the possible style(Luong et al., 2015)(John et al., 2018)(Balikas et al., 2017).

# 4 Experiments

## 4.1 Dataset

We collect the dataset from a social media platform *REDBook* with plenty of life-sharing "hot post" records. As long as the content/headline is compelling enough, the blog will get more exposure and attract more followers. "Likes" is a measure of a post's popularity, which is also proof of eye-catchy quality. So we filter 70k different posts with more than 2k "likes", shown in Table 1 for the headline generator training period. To extract the global contextual features, we select the content and corresponding headline of hot post, from

different bloggers with the consideration of avoiding personal-style influence. We randomly divide the *REDBook* dataset into train set, validation set, and test set. For the inference and selection of the best candidate task, we randomly choose 20k posts (RED-IN). Our dataset contains the hot post published during 2021, and time is not an influencing factor as the like counts have already accumulated.

| Dataset | Size | Document | | Title |
|---|---|---|---|---|
| | | Avg-Len | Avg-Sens | Avg-Len |
| **RED-Tr** | 70k | 148.5 | 11.7 | 16.4 |
| **RED-In** | 20k | 149.4 | 11.9 | 16.4 |

Table 1: Different *REDBook* datasets are used for HG generation training and budgets constraint inference time, respectively.

## 4.2 Baselines

We select the related Seq2seq summarization methods:*BART*(Lewis et al., 2019) and *PEGASUS*(Zhang et al., 2020) as basic large pre-trained standard in the literature. The tokenizers of BART and PEGASUS are also well-established with Chinese datasets.

**Implementation Details** In the following experiments, we use either *BART* or *PEGASUS* as a backbone. We label our proposed methods **MediaHG** with several variants: (1) **MediaHG-BA** is fine-tuned with eye-catchy features based on *BART*. (2)**MediaHG(-PG)** is fine-tuned with eye-catchy features based on *PEGASUS*. We also change the eye-catchy features influence as (3) **MediaHG-c** using content constraint only and **MediaHG-s** using style constraint only. The choice of sample times $k$ also has a great influence on the results. So we set (4) **MediaHG-m** with $k = 10$.

## 4.3 Experiment Settings

Consistent with the platform requirements, we set the maximum target title length as 20 characters for all models. According to the average length of the documents and titles, we set the max length of the tokenizer as 512. The encoder and the decoder of all Seq2Seq models are set as the same parameters as *BART*(Lewis et al., 2019) and *PEGASUS*(Zhang et al., 2020). As we have discussed in Sec 3.1, the number of sentences in document views $S_v$ is controlled by the sampling factor parameter $s_f = \frac{2}{3}$. Another hyperparameter $k$ to control the number of samples $|S_v|$ is set as 5 and 10. During the inference period with content and style

budgets to select the best headline, the eye-catchy feature budgets are respectively set with $\alpha = 0.5$ and $\gamma = 0.1$.

### 4.4 Evaluation Metrics

**ROUGE:** We evaluate models using standard full-length ROUGE F1 (Lin, 2004) following previous works(Li et al., 2021)(Gao et al., 2019)(Xu et al., 2019). ROUGE-1, ROUGE-2 and ROUGE-L refer to the matches of unigrams, bigrams, and the longest common subsequence, respectively.

**BLEU:** To evaluate our model more comprehensively, we also use the metric BLEU proposed by (Papineni et al., 2002) which measures word overlap between the generated text and the ground-truth.

**Human Evaluation:** As single autometric evaluation can be misleading (Schluter, 2017), we add human evaluation metrics to our work. We randomly sample 500 cases from the test set and ask three familiar and loyal *REDBook* users as annotators to score the headlines generated by *BART*, *PEAGUSUS*, and *MediaHG*. Referring to the gender and age distribution of *REDBook* users, reviewers consist of a man and two women about 30 years old.

### 4.5 Results

**Overall Performance** We compare our model with baselines in Table 2. Firstly, PEGASUS still outperforms BART, which means our task needs a bit more abstractive summarization. Secondly, our model achieves 21.46, 7.79, 19.05, and 11.26 in terms of ROUGE-1, ROUGE-2, and ROUGE-L respectively outperforming both PEAGASUS and BART and thus proves the superiority of our model. Besides, *MediaHG* outperforms *MediaHG-BA* in terms of all the metrics scores. An example of headlines generated by *BART*, *PEAGASUS*, and our model *MediaHG* can be found in Appendix B. We refer readers to Appendix B for more details.

We also add *MediaHG-c* and *MediaHG-s* to see the eye-catchy features set influence. *MediaHG-c* achieves better automatic scores than *MediaHG-s*, which means content budget impact on the main topic extraction of the headline. The disparity between *MediaHG* and *Media-c* illustrates the style-content duality of a good eye-catchy headline.

The *MediaHG-m* model shares the same parameters with *MediaHG* except the number of sample times $k$. The outperformance of *MediaHG-m* also certificates the importance of sampling. However,

| Model | R-1 | R-2 | R-L | BLEU |
|---|---|---|---|---|
| BART | 18.26 | 6.98 | 16.26 | 7.49 |
| PEGASUS | 20.05 | 7.17 | 17.83 | 10.33 |
| MediaHG-BA | 18.50 | 5.75 | 16.61 | 9.84 |
| MediaHG-c | 20.85 | 7.08 | 18.68 | 11.64 |
| MediaHG-s | 20.43 | 7.48 | 17.92 | 11.16 |
| **MediaHG(-PG)** | **21.46** | **7.79** | **19.05** | **11.26** |
| **MediaHG-m** | **23.66** | **8.70** | **20.78** | **12.30** |

Table 2: Results of our models MediaHG and baselines on each of the automatic evaluation metrics. MediaHG-BA uses MediaHG inference methods based on BART as MediaHG(-PG) based on PEGASUS. MediaHG-c and MediaHG-s are constrained with single content and style budgets. MediaHG-m uses $k$ as 10 while other parameters are the same as MediaHG(-PG). R-1/2/L are the ROUGE-1/2/L $F_1$ scores.

as the number of sampling increases, experiment time increases correspondingly.

The human evaluation is based on 3 aspects: sentence fluency, content faithfulness, and contextual eye-catchy requirements. The rating score of each model ranges from 1 to 3, with 3 being the best. Additionally, headlines with domain features like female tongue will get better scores in attractiveness. Table 3 lists the average scores of each model, demonstrating that *MediaHG* outperforms other baseline models.

| | Flu | Con | Attr |
|---|---|---|---|
| BART | 2.05 | 2.13 | 1.76 |
| PEGASUS | 2.24 | 2.02 | 2.03 |
| **MediaHG** | **2.54** | **2.37** | **2.43** |

Table 3: Fluency(Flu), consistency(Con) and attractiveness(Attr) comparison by human evaluation. The data is the average score of the three labelers' results.

### 4.6 Analysis

We further analyze the contribution of different module parts from diverse perspectives to gain more insights into our method.

**Coefficients of Contrastive Loss** The global training loss contains two parts: the cross-entropy

| | R-1 | R-2 | R-L | BLEU |
|---|---|---|---|---|
| **MediaHG** | 21.46 | 7.79 | 19.05 | 11.26 |
| **MediaHG-lo** | **22.05** | **7.96** | **20.14** | **11.87** |

Table 4: Comparison of results from MediaHG and fine-tuned twice generation model MediaHG-lo. R-1/2/L are the ROUGE-1/2/L F1 scores

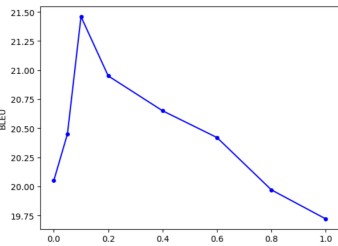

Figure 4: Model performance with different $\gamma$ coefficients weighting the contrastive loss(Eq.7).

| $s_f$ | R-1 | R-2 | R-L | BLEU |
|-----|-------|------|-------|-------|
| 1/2 | 21.25 | 7.65 | 18.92 | **11.35** |
| 2/3 | **21.45** | 7.78 | 19.05 | 11.26 |
| 3/4 | 21.43 | **7.93** | **19.10** | 11.28 |
| 1 | 20.05 | 7.17 | 17.83 | 10.33 |

Table 5: Results with different choice of sample size $S_f$.R-1/2/L are ROUGR-1/2/L $F_1$ scores.

loss and the contrastive loss. In order to study the influence of the contrastive learning module, we train our model with different contrastive learning coefficients $\gamma$. As the cross-entropy loss is necessary to predict sequential tokens and preserve the generation model ability, we only change the values of $\gamma$ (shown in Figure 4). When $\gamma$ is smaller than 0.1, the larger $\gamma$, the better of performance. But when $\gamma$ is bigger than 0.1, the smaller $\gamma$ leads to better performance. When $\gamma$ is small, the contrastive learning module has a little positive impact on the whole model, so the results are getting better. While the $\gamma$ increases, contrastive impact too much on the whole model, which disturbs the training of headline generation.

**Generation-Fintune as a Loop** As our inference selection results are style-content dual, a new set of candidates can be generated in the same way as the pre-trained model dynamically and continuously. Table 4 illustrates the effectiveness of this loop operation. It also demonstrates our method's potential improvement in headline generation.

### 4.7 Ablation Study

In order to verify the effect of each module in MediaHG, we conduct ablation tests in Table 5 and Table 6. As we have discussed, the number of sentences in document views $S_v$ is controlled by the sampling factor parameter $s_f \in [0, 1]$, so we choose $\frac{1}{2}, \frac{2}{3}, \frac{3}{4}$ to perform experiments respectively shown in Table 6. Different sampling size brings different best scores, but all outperform scores with

| k | R-1 | R-2 | R-L | BLEU |
|----|-------|------|-------|-------|
| 4 | 20.58 | 7.46 | 18.41 | 10.83 |
| **5** | **21.46** | **7.79** | **19.05** | **11.26** |
| 6 | 21.89 | 7.96 | 19.38 | 11.47 |
| 7 | 22.56 | 8.25 | 19.85 | 11.75 |
| 8 | 23.07 | 8.46 | 20.37 | 12.00 |
| 9 | 23.23 | 8.44 | 20.40 | 12.10 |
| **10** | **23.66** | **8.70** | **20.78** | **12.30** |

Table 6: Results with different sample number of times k.R-1/2/L are ROUGR-1/2/L $F_1$ scores.

$s_f = 1$. The results indicate the necessity of sampling in extracting the main topic.

Another hyperparameter **k** to control the number of samples $|S_v|$ is set from 4 to 10 due to the actual needs. We see a rapid increase from 4 to 5 and then a slower increase. As the **k** rises, more selections are given to the selector, so the scores will increase accordingly. With the dataset length features (Table 1), we set the max **k** as 10 to create various but nonredundant results. At the same time, for the level of higher complexity, the experimental time increases much.

## 5 Conclusion and Future

In this paper, we propose an eye-catchy headline generation model *MediaHG* for vertical interest social media platforms. Our research is the first one focusing on vertical interest websites. As people's interests flourish with the information gap broken down, more websites will be designed to appeal to the same interest groups. Our design allows the features extractor approach to be used more flexibly with other websites' data. Both automatic and human evaluation show our improvement in headline generation.

## Limitations

When dealing with texts of different lengths, selecting parts and generating headlines may result in redundant similar candidates or insufficient information. It is necessary to select appropriate model parameters according to the characteristics of posts.

## Acknowledgements

This work was supported by NSFC (62322606, 62176233) and the National Key Research and Development Project of China (2018AAA0101900).

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

**Algorithm 1:** Candidates Selection. Input parameters are the candidate headlines $\mathcal{H}$, weight of dual balance $\alpha$, content measure method $m_c$ and style measure $m_s$, size of beam search $b_1, b_2$. The output $H^*$ is the best headline after selection.

**Input:** $\mathcal{H}, \alpha, m_c, m_s, b_1, b_2$
**Output:** $H^*$
1  initialization
2  $H^* \leftarrow \emptyset, i \leftarrow 0$
3  $S_c \leftarrow \emptyset, S_v \leftarrow \emptyset$
4  $\mathcal{M}(H_i) = \alpha m_c(H_i) + (1-\alpha)m_s(H_i)$
5  **for** $H_i \in \mathcal{H}$ **do**
6  $\quad \mid \quad S_c \leftarrow S_c \cup \{\, \langle H_i, m_c(H_i) \rangle \}$
7  **end**
8  $B \leftarrow S_c.top(b_1)$
9  **for** $H_i \in B$ **do**
10 $\quad \mid \quad S_s \leftarrow S_s \cup \{\, \langle H_i, m_s(H_i) \rangle \}$
11 **end**
12 $C \leftarrow S_s.top(b_2)$
13 $\hat{\mathcal{H}} \leftarrow \mathcal{M}(H_i \in C)$
14 $H^* \leftarrow \hat{\mathcal{H}}.max$

# A  Appendix

## A.1  Algorithm 1

An accurate prediction of the best headline ranking procedure is required before output. To avoid big waste in enumerating all possible candidates, we offer a beam search algorithm, especially suitable for plenty of candidates' tasks.

## A.2  Case Study

We display an example of headlines generated by BART, PEAGASUS, and our model MediaHG in Table 7. The headline generated by MediaHG is more informative and attractive compared with the BART model and PEGASUS model. We also show the document of the hot post to certify the content consistency of the generated headlines.

The baselines can generate fluent headlines in this case, but they miss the attractive style and will include unattractive content. The headline generated by BART is a plain statement, neglecting the main attractive topic "green eyeliner painting". It only catches the word "hottie", which does not fully encapsulate the character of the article. We also consider the faithfulness of different models. The headline generated by PEGASUS is significantly better than BART, as it catches the main word "green eyeliner painting"(in blue). But headlines generated with inappropriate elements still exist as we highlight in red. For our model in the case, MediaHG captures the keywords "green eyeliner" and "hot girl", which not only cover the topic but also draw more attention. This case also demonstrates our sampling module effects in main topic extraction.

| | |
|---|---|
| **English Document** | Dear Family Fans, I'm here with a new recreation to unlock the green eyeliner! Please, try it, you can't miss it! The whole concept of the look is "Forcing to add a little bit of green on this freezing cold day". It's a sign of my determination to be a hot spice girl! Give it a shot! Spice Girl Step 1 Hybrid con: MITATA contact lens. A little bit of a blue-brown gradation makes for a very artistic look. It's also quite premium!! Spice Girl Step 2 Hairstyle: I also unlock the double-balls hairstyle today. Very cute, with an American school-style feeling! Spice Girl Step 3 Unique Eye Makeup: the green color of this palette called "huazhixiao"02, is so wonderful, right? Use a small brush and dip in green color to draw eyeliner, dotted with glowing sequins in the crowd! nice! |
| **Chinese Document** | 家人们，我带着新活来了，解锁了绿色眼线画法!请家人们品品,这能不试试看嘛!整个妆容的概念就是"强行在这个冷到我哆嗦的天里，加一点绿",代表我要做辣妹的斗志和决心！给我冲！辣妹第一步混血con: MITATA美瞳有一点蓝褐色的渐变很有艺术感，也很高级绝了！！辣妹第二步发型:今天还解锁了双丸子头，很cute了有点美式校园风的感觉。辣妹第三步独特的眼妆:花知晓02这盘的绿色也太好了吧，用小刷子沾取绿色画一条眼线，点上亮片人群中在发光! nice！ |
| **Reference Title** | California Hotties!! Green Eyeliner really looks Pretty! 加州辣妹！！绿色眼线真的好好看! |
| **BART** | Hottie, European and American Makeup Style, Hot! 辣妹辣妹，欧美妆容，辣！ |
| **PEGASUS** | European and American Style Makeup, Green Eyeliner Painting Methods, Hot Girl's Fighting Spirit and Determination. 欧美妆容，绿色眼线画法，辣妹的斗志和决心 |
| **MediaHG** | European and American Style Makeup, Green Eyeliner Painting Methods, Hot Girls' Good Looking! 欧美妆容，绿色眼线画法，辣妹好看！ |

Table 7: Case study to verify the effectiveness of MediaHG. To reveal the influence of sampling on style and content space, we highlight the comparison between the document and title. The sentence in blue denotes attractive information related to the reference headline and document, and the text in red denotes information related to the document but not proper for a headline.