# OpenReview forum: "MediaHG: Rethinking Eye-catchy Features in Social Media Headline Generation"
_EMNLP/2023/Conference — EMNLP 2023 Main_

### Official Review · Reviewer_r8GS · 2023-07-20

**Soundness:** 4

**Excitement:**

4: Strong: This paper deepens the understanding of some phenomenon or lowers the barriers to an existing research direction.

**Paper Topic And Main Contributions:**

The paper introduces "MediaHG," a disentanglement-based model designed for social media headline generation. This model effectively balances between content and contextual attributes. It is developed using data from 70,000 popular posts on REDBook, a renowned Chinese social media platform. Evaluations using the ROUGE metric and human assessment reveal significant advancements in headline generation for the platform.

**Reasons To Accept:**

1. The paper addresses a significant problem in the realm of social media, focusing on the balance between content relevance and the appeal of headlines. The paper takes into account the uniqueness of platform features, catering to specific needs of different websites.

2.  The clear delineation between the generation and selection process is commendable. By separating these two steps, the model allows for the possibility to reuse the ranking process in different scenarios. This modular approach could be instrumental in adapting to various use-cases or platforms.

3. The paper conducts human evaluation and provides a detailed case study. By doing so, the authors provide tangible evidence of the efficacy and quality of the generated headlines.

**Reasons To Reject:**

1. The paper does not clarify whether the dataset used, sourced from REDBook, will be made available for public use. A publicly accessible dataset would benefit the broader research community, ensuring reproducibility and facilitating comparison with other models. Without this, it becomes challenging for peers to reproduce or validate the study's findings.

2. The idea of bifurcating content and style resembles the approach described in "The style-content duality of attractiveness: Learning to write eye-catching headlines via disentanglement." The current paper would benefit from a deeper, more detailed comparison with this prior work rather than a brief mention or superficial comparison.

3. While the paper claims theoretical flexibility for the MediaHG model, there's a noticeable absence of empirical backing. To substantiate this claim, it would be beneficial to test the model on multiple datasets, ensuring a comprehensive evaluation and lending more credibility to the notion of its adaptability.

**Reproducibility:**

3: Could reproduce the results with some difficulty. The settings of parameters are underspecified or subjectively determined; the training/evaluation data are not widely available.

**Reviewer Confidence:**

4: Quite sure. I tried to check the important points carefully. It's unlikely, though conceivable, that I missed something that should affect my ratings.

---

> ### Author Rebuttal · Authors · 2023-08-29
>
> We greatly appreciate your positive review and valuable feedback, and we are committed to addressing your concerns fully in our response.
>
> **W1:** The paper dose not clarify whether the dataset will be available for pulic use.
>
> **A1:** Our work is the first to generate topic-catchy and contextually harmonized headlines for vertical niche platforms, aiming to enhance users' experience. In consideration of privacy protection, we will make our dataset available after undergoing a thorough data cleaning process. This will benefit broader research areas such as niche platform preference and stylistic headline generation. We also welcome any attempts to reproduce or validate our findings.
>
> **W2:** The paper would benefit from a deeper comparison with "The Style-Content Duality of Attractiveness: Learning to Write Eye-Catching Headlines via Disentanglement".
>
> **A2:** In the mentioned work[5], disentanglement is used to separate the style and content into latent spaces, but the extractor requires an additional language training process. To make our model more flexible and reduce computational complexity, we generate a set of candidates and select the most suitable one simultaneously. In fact, our model introduces a novel approach in constructing the style and content space constraints, with reranking candidates generated by the Seq2Seq model.
>
> **W3:** The paper lacks test on multiple datasets.
>
> **A3:** Our method, MediaHG, is the first work focused on vertical niche platforms to improve the click ratio and attract users’ attention. Hence, it is challenging to find similar publicly available datasets with the same background characteristics. We attempt to find related stylist headline generation work, such as “The Style-Content Duality of Attractiveness: Learning to Write Eye-Catching Headlines via Disentanglement”[5], but the link of Kuaibao[6] dataset mentioned in the paper is currently inaccessible. Another dataset, Chinese LSCC news corpus, is also unavailable in recent work,"SCS-VAE: Generate Style Headlines Via Novel Disentanglement"[8].
>
> Due to the lack of public available datasets with similar background, we finally find another headline generation work called “Hooks in the Headline: Learning to Generate Headlines with Controlled Styles" (TitleStylist)[1], which uses the classic summarization dataset CNN[2]. To ensure the robustness and applicability of MediaHG, we compare the results of various models on the CNN dataset and list them as below:
>
> |         | R-1   | R-2   | R-L   |
> | ------- | ----- | ----- | ----- |
> | BART    | 44.16 | 21.28 | 40.90 |
> | PEGASUS | 44.17 | 21.47 | 41.11 |
> |MediaHG|**46.74**|**22.78**|**42.84**|
>
> Since the REDBook dataset is in Chinese, our MediaHG is focused on Chineses corpus summarization using PEGASUS[4] (and BART[3]). As a result, MediaHG performs better than the baseline models BART and PEGASUS on the supplementary selection part. The headline generator part of MediaHG can be replaced with other Seq2Seq models based on multiple datasets with different languages or backgrounds.
>
> We sincerely hope this explanation addresses your concerns. If you come across any similar datasets, please feel free to recommend them to us!
>
> **[Reference]**
>
> [1] Hooks in the headline: Learning to generate headlines with controlled styles.
>
> [2] Teaching machines to read and comprehend.
>
> [3] BART: Denoising sequence-to-sequence pretraining for natural language generation, translation, and comprehension.
>
> [4] Pegasus: Pre-training with extracted gap-sentences for abstractive summarization.
>
> [5] The style-content duality of attractiveness: Learning to write eye-catching headlines via disentanglement.
>
> [6] Automatic article commenting: the task and dataset.
>
> [7] Mass: Masked sequence to sequence pre-training for language generation.
>
> [8] SCS-VAE: Generate Style Headlines Via Novel Disentanglement.
>
> Again, we appreciate your valuable reviews and hope our response can fully resolve your concerns.

---

### Official Review · Reviewer_miJc · 2023-08-03

**Soundness:** 4

**Excitement:**

3: Ambivalent: It has merits (e.g., it reports state-of-the-art results, the idea is nice), but there are key weaknesses (e.g., it describes incremental work), and it can significantly benefit from another round of revision. However, I won't object to accepting it if my co-reviewers champion it.

**Paper Topic And Main Contributions:**

This paper focus on generating headlines for social media users’ posts to promote the users’ interaction. To obtain attractive headlines, the authors propose a MediaHG model, which first randomly combines sentences to generate headline candidates and then uses a rerank module to select the best title based on the style and content. Additionally, a dataset that contains popular posts is collected. Experimental results with automatic metrics and human evaluation on the collected dataset show the effectiveness of the proposed model.

**Reasons To Accept:**

1. The paper proposes a novel MediaHG model to generate eye-catchy headlines to enhance user interaction, where a sample module is designed to obtain different documents views, and contrastive learning and multi-task learning are utilized to choose the best sentence among the headline candidates generated from the headline generator.
2. Authors collect a headline generation dataset REDBook, which has the potential to foster community development.
3. The designed modules are general and could be employed on other models. Experimental results indicate the effectiveness of the proposed modules.


**Reasons To Reject:**

1. The paper lacks sufficient information regarding the data collection process, which raises concerns about potential ethical issues. To ensure transparency and address any ethical considerations, it is essential for the authors to provide detailed insights into their data collection methodology and how they handled any ethical implications that may have arisen during the process.
2. The absence of a case study to showcase the quality of headlines generated by different models is a limitation of the paper.
3. The human evaluation result of MediaHG-BA should be included in Table 3.


**Reproducibility:**

3: Could reproduce the results with some difficulty. The settings of parameters are underspecified or subjectively determined; the training/evaluation data are not widely available.

**Reviewer Confidence:**

3: Pretty sure, but there's a chance I missed something. Although I have a good feel for this area in general, I did not carefully check the paper's details, e.g., the math, experimental design, or novelty.

---

> ### Author Rebuttal · Authors · 2023-08-29
>
> We appreciate the reviewer's constructive feedback and their recognition of our work's potential to foster community development and have a positive impact on others. We understand that the reviewer's concerns primarily revolve around the ethical aspects of our paper, and we aim to address these concerns comprehensively in our response.
>
> **W1:** The paper lacks sufficient information regarding the data collection process.
>
> **A1:** To obtain the data for our study, we utilized web scraping techniques on the publicly available information from the REDBook website. The "public information" refers to content that is visible to all platform users. The data we scraped is referred to as the raw dataset, which consists of 1,695,219 posts and 42,447 users. Each post included various features such as the headline, content, likes count, and writer. In order to construct our experimental dataset for eye-catchy detection, we filtered out 70k distinct posts with more than 2k likes. To ensure a diverse selection and avoid personal-style bias, we selected posts in a balanced manner from different bloggers.
>
> **W2:** The absence of a case study to showcase the quality of different models.
>
> **A2:** We appreciate the reviewer's feedback regarding the case study in our work. Due to the constraints of the paper's length, we have included the case study in Appendix B and mentioned it in Section 4.5. The case study presents the results in two languages (English and Chinese) from three models for comparison, with MediaHG achieving the best performance according to the reference. We hope that the inclusion of the case study comparison adequately addresses the reviewer's concerns, and we welcome any further advice they may have.
>
> **W3:** The human evaluation result of MediaHG_BA should be included in Table 3.
>
> **A3:** In our evaluation, we employed human evaluation from three aspects: sentence fluency, content faithfulness, and contextual eye-catchy requirements, specifically assessing the performance of the generated headlines. To highlight the advantages of our design, we compared the performance of MediaHG(-PG) with two baseline models (BART and PEGASUS). Additionally, we introduce an additional candidate selection framework for BART, referred to as MediaHG-BA, which also demonstrates the benefits of our approach. We invite the same evaluators as before to assess the results from MediaHG-BA, and the average scores obtained were as follows: sentence fluency: 2.01, content faithfulness: 2.13, and contextual eye-catching requirements: 1.87.
>
> We include the MediaHG-BA results in Table 3, which are presented as follows:
>
> |         | Flu  | Con  | Attr |
> |:-----:|:------:|:------:|:------:|
> | BART    | 2.05 | 2.13 | 1.76 |
> | PEGASUS | 2.24 | 2.02 | 2.03 |
> |MediaHG-BA|2.17|2.13|1.87|
> |MediaHG(-PG)|**2.54**|**2.37**|**2.43**|
>
> Again, we appreciate your valuable reviews and hope our response can fully resolve your concerns.

---

### Official Review · Reviewer_zt61 · 2023-08-05

**Soundness:** 3

**Excitement:**

4: Strong: This paper deepens the understanding of some phenomenon or lowers the barriers to an existing research direction.

**Paper Topic And Main Contributions:**

This paper presents MediaHG, a novel disentanglement-based headline generation model that adeptly balances content and contextual features to craft headlines that are fitting for their respective domains. The paper addresses the challenge of creating compelling and informative headlines for social media posts. The primary contributions of the paper are threefold:

1. The introduction of a disentanglement-based model, MediaHG, which effectively separates content and contextual features, enabling the generation of domain-appropriate headlines.
2. The development of a new dataset, REDBook, specifically designed for the evaluation of headline generation models.
3. The empirical demonstration of MediaHG's effectiveness using the REDBook dataset, with comparative analysis against other state-of-the-art models, underscoring its superior performance.


**Reasons To Accept:**

The paper presents MediaHG, a novel disentanglement-based headline generation model that adeptly balances content and contextual features to craft headlines that are fitting for their respective domains. The strengths of this paper are manifold:

1. The introduction of a unique disentanglement-based model for headline generation, which effectively separates content and contextual features, enabling the generation of domain-appropriate headlines. This novel approach could be applied to other domains beyond social media, expanding its potential impact.
2. The development of a new dataset, REDBook, specifically designed for the evaluation of headline generation models. This dataset enriches the resources available to the NLP community for research in this area.
3. The empirical demonstration of MediaHG's effectiveness using the REDBook dataset, with comparative analysis against other state-of-the-art models. This rigorous evaluation not only underscores the superior performance of MediaHG but also sets a benchmark for future research in this area.
4. The potential application of the proposed model to improve the quality of headlines generated for social media posts. This could benefit both content creators and readers by enhancing the informativeness and engagement of social media content.


**Reasons To Reject:**

The paper presents MediaHG, a novel disentanglement-based headline generation model that balances content and contextual features to create domain-suitable headlines. While the paper makes significant contributions, it also has potential weaknesses:
1. The proposed model is evaluated on a single dataset, REDBook. This may limit the generalizability of the results to other social media platforms or domains. Future work could involve testing the model on a variety of datasets to ensure its robustness and applicability across different contexts.
2. The paper does not provide a thorough comparison with other state-of-the-art models for headline generation. This may limit the ability to assess the effectiveness of the proposed model relative to other approaches. Future work could involve a more comprehensive comparison with other models, considering various metrics and scenarios.
3. The paper does not provide a detailed analysis of the limitations of the proposed model or potential areas for future research. Future work could involve a more in-depth discussion on these aspects, providing a more balanced view of the model and paving the way for further improvements.
4. The potential misuse of the proposed model for creating clickbait or for other unethical purposes is not discussed in the paper. Future work should consider these ethical implications and possibly propose mechanisms to prevent such misuse.


**Reproducibility:**

3: Could reproduce the results with some difficulty. The settings of parameters are underspecified or subjectively determined; the training/evaluation data are not widely available.

**Reviewer Confidence:**

4: Quite sure. I tried to check the important points carefully. It's unlikely, though conceivable, that I missed something that should affect my ratings.

---

> ### Author Rebuttal · Authors · 2023-08-29
>
> We are grateful for your positive review and valuable feedback, and we hope our response fully resolves your concern.
>
> **W1:** The proposed model is evaluated on sindle dataset.
>
> **A1:** Our method, MediaHG, is the first work focused on vertical niche platforms to improve the click ratio and attract users’ attention. Hence, it is challenging to find similar publicly available datasets with the same background characteristics. We attempt to find related stylist headline generation work, such as “The Style-Content Duality of Attractiveness: Learning to Write Eye-Catching Headlines via Disentanglement”[5], but the link of Kuaibao[6] dataset mentioned in the paper is currently inaccessible. Another dataset, Chinese LSCC news corpus, is also unavailable in recent work,"SCS-VAE: Generate Style Headlines Via Novel Disentanglement"[8].
>
> Due to the lack of public available datasets with similar background, we finally find another headline generation work called “Hooks in the Headline: Learning to Generate Headlines with Controlled Styles" (TitleStylist)[1], which uses the classic summarization dataset CNN[2]. To ensure the robustness and applicability of MediaHG, we compare the results of various models on the CNN dataset and list them as below:
>
> |         | R-1   | R-2   | R-L   |
> | ------- | ----- | ----- | ----- |
> | BART    | 44.16 | 21.28 | 40.90 |
> | PEGASUS | 44.17 | 21.47 | 41.11 |
> |MediaHG|**46.74**|**22.78**|**42.84**|
>
> Since the REDBook dataset is in Chinese, our MediaHG is focused on Chineses corpus summarization using PEGASUS[4] (and BART[3]). As a result, MediaHG performs better than the baseline models BART and PEGASUS on the supplementary selection part. The headline generator part of MediaHG can be replaced with other Seq2Seq models based on multiple datasets with different languages or backgrounds.
>
> We sincerely hope this explanation addresses your concerns. If you come across any similar datasets, please feel free to recommend them to us!
>
> **W2:** The paper lacks thorough comparison with other SOTA models.
>
> **A2:** We attempt to apply other stylistic headline generation models to the REDBook dataset but find them to be inapplicable. Our model consists of 3 parts:  *Headline Generator* ,*Sample Module* and *Features Reranking*, all trained with REDBook dataset. The input to the reranking part is the output of the headline generator part. For flexibility, our headline generator is based on Seq2Seq summarization models BART and Pegasus to generate headlines candidates. The related previous work TitleStylist[1] requires additional clickbait corpus, SpamClickBait News, to train the clickbait features. It also employs a Seq2Seq architecture[7] to generate three styles (Humor, Romance, and Clickbait), but we only refer to the Clickbait generation. Since it is difficult to find additional Clickbait corpora in Chinese, we use the CNN[2] dataset in TitleStylist for result comparison. The results are as follows:
>
> |         | R-1   | R-2   | R-L   |
> | ------- | ----- | ----- | ----- |
> |TitleStylist|26.6|9.8|23.7|
> |MediaHG|**46.74**|**22.78**|**42.84**|
>
> We find a significant gap in results between TitleStylist and MediaHG, which may be attributed to the underlying differences in Seq2Seq models. Consistency in training data across different parts may also contribute to the results as it enhances content relevance.
>
> We also compared our work with other headline generation models on the CNN[2] dataset in the last part 1.
>
> **W3:** The paper shall provide detailed analysis for future work.
>
> **A3:** In the paper, we have mentioned two limitations: parameter selection and clickbait criteria. The parameters may be constrained by our single dataset with specific sentence lengths, headline lengths, characteristic counts, etc. The eye-catchy feature is filtered based on likes, which may be incomplete. Therefore, we suggest considering additional features or human subjective factors for evaluation. As manual selection requires a significant amount of time, we hope future creative work can provide a precise yet convenient evaluation source.
>
> **W4:** The potential misuse of clickbait is not discussed.
>
> **A4:** To avoid solely focusing on clickbait, we prioritize **Rouge** scores to indicate the relevance between generated headlines and corresponding posts in our paper. During the candidate selection process, we also assign separate scores for content and contextual style. The Rouge score can serve as a detection strategy to prevent the misuse of clickbait.
>
> **[Reference]**
>
> [1] Hooks in the headline: Learning to generate headlines with controlled styles.
>
> [2] Teaching machines to read and comprehend.
>
> [3] BART: Denoising sequence-to-sequence pretraining for natural language generation, translation, and comprehension.
>
> [4] Pegasus: Pre-training with extracted gap-sentences for abstractive summarization.
>
> [5] The style-content duality of attractiveness: Learning to write eye-catching headlines via disentanglement.
>
> [6] Automatic article commenting: the task and dataset.
>
> [7] Mass: Masked sequence to sequence pre-training for language generation.
>
> [8] SCS-VAE: Generate Style Headlines Via Novel Disentanglement.
>
> Again, we appreciate the reviewer's valuable reviews. We hope our response can address your concerns.

---

### Meta-Review · Area_Chair_wDpZ · 2023-09-07

**Recommendation:** 5

**Metareview:**

This paper proposes a disentanglement-based headline generation model for social media that balances content (i.e. capturing the main topic of the document) and contextual features (i.e. producing "eye-catchy" headlines). Overall, reviewers are quite positive about this work and I feel that the paper's strengths (e.g. sound and effective model, introduced dataset, application beyond social media) outweigh its weaknesses (e.g. lack of details about the proposed dataset, limited scope of the experiments). It should be noted that some of these concerns were discussed and, I believe, addressed by the author's responses.

I would also like to see the potential ethical issues related to this work thoroughly discussed in the paper (i.e. misuse of the model for click-bait (#R zt61), sourcing of the REDBook dataset (#R r8GS and miJc)).

---

### Decision · Program_Chairs · 2023-10-07

**Decision:**

Accept-Main

**Comment:**

This paper proposes a disentanglement-based headline generation model for social media that balances content (i.e. capturing the main topic of the document) and contextual features (i.e. producing "eye-catchy" headlines). Overall, reviewers are quite positive about this work and I feel that the paper's strengths (e.g. sound and effective model, introduced dataset, application beyond social media) outweigh its weaknesses (e.g. lack of details about the proposed dataset, limited scope of the experiments). It should be noted that some of these concerns were discussed and, I believe, addressed by the author's responses.

I would also like to see the potential ethical issues related to this work thoroughly discussed in the paper (i.e. misuse of the model for click-bait (#R zt61), sourcing of the REDBook dataset (#R r8GS and miJc)).